# Targeting Nrf2 with Probiotics and Postbiotics in the Treatment of Periodontitis

**DOI:** 10.3390/biom12050729

**Published:** 2022-05-22

**Authors:** Basar Karaca, Mustafa Yilmaz, Ulvi Kahraman Gursoy

**Affiliations:** 1Department of Periodontology, Institute of Dentistry, University of Turku, 20520 Turku, Finland; mustafa.yilmaz@utu.fi; 2Department of Biology, Faculty of Science, Ankara University, Ankara 06100, Turkey; 3Department of Periodontology, Faculty of Dentistry, Biruni University, Istanbul 34010, Turkey

**Keywords:** antioxidant, lactic acid bacteria (LAB), Nrf2, oxidative stress, periodontitis, postbiotics, probiotics

## Abstract

Periodontitis is a destructive disease of the tooth-surrounding tissues. Infection is the etiological cause of the disease, but its extent and severity depend on the immune–inflammatory response of the host. Immune cells use reactive oxygen species to suppress infections, and there is homeostasis between oxidative and antioxidant mechanisms during periodontal health. During periodontitis, however, increased oxidative stress triggers tissue damage, either directly by activating apoptosis and DNA damage or indirectly by activating proteolytic cascades. Periodontal treatment aims to maintain an infection and inflammation-free zone and, in some cases, regenerate lost tissues. Although mechanical disruption of the oral biofilm is an indispensable part of periodontal treatment, adjunctive measures, such as antibiotics or anti-inflammatory medications, are also frequently used, especially in patients with suppressed immune responses. Recent studies have shown that probiotics activate antioxidant mechanisms and can suppress extensive oxidative stress via their ability to activate nuclear factor erythroid 2-related factor 2 (Nrf2). The aim of this narrative review is to describe the essential role of Nrf2 in the maintenance of periodontal health and to propose possible mechanisms to restore the impaired Nrf2 response in periodontitis, with the aid of probiotic and postbiotics.

Periodontitis is a highly prevalent inflammatory disease of an infectious origin, and it is the major cause of tooth loss globally. Increased oxidative stress and diminished antioxidant capacity are associated with periodontal pathogenesis. While conventional periodontal treatment is effective in many individuals, some unresponsive patients can theoretically benefit from adjunctive therapies that may have both direct effects on biofilm control and indirect effects, such as modulation of the host response. Regulation of antioxidant capacity with adjunctive therapy models is an emerging topic in various medical fields, particularly via nuclear factor erythroid 2-related factor 2 (Nrf2) activation. Nrf2 is a transcription factor that has been the subject of numerous studies, revealing the mechanism behind the oxidative-stress-induced cellular damage. This narrative review aimed to describe the Nrf2 activating abilities of postbiotics and address the potential benefit of postbiotic use as an adjunct to conventional periodontal treatment.

## 1. Periodontal Health and Disease

The periodontium consists of four specialized tissues that work together harmoniously: Alveolar process (bone that forms the walls of the tooth socket), periodontal ligament (connective tissue that joins the tooth to the alveolar bone), cementum (thin hard tissue that lines the root surface where the fibers of the periodontal ligament are inserted), and gingiva (oral mucosa that abuts the tooth and seals the underlying tissues). Periodontal health is defined by the absence of clinical signs of gingival inflammation and alveolar bone loss [1]. However, even in clinically healthy gingiva, periodontal tissues resist constant microbial challenges through innate and adaptive defense mechanisms [2]. In susceptible individuals, the balance between the subgingival microflora and the host response is disturbed, resulting in uncontrolled, persistent inflammation and consequent loss of alveolar bone and periodontium, namely periodontitis [3,4].

Periodontitis is an infectious inflammatory disease. Clinical signs of periodontitis include gingival inflammatory changes, such as redness, swelling, bleeding, pocket formation and/or gingival recession, bone loss, increased tooth mobility, pathologic tooth migration, suppuration, and abscess formation, and it can lead to tooth loss (Figure 1). In addition to oral consequences, periodontitis is also associated with various systemic diseases and conditions, such as hyperglycemia, chronic pulmonary diseases, cardiovascular disease, and obstetric complications, which may be related to periodontitis and the systemic consequences of chronic, low-grade inflammation [5].

Periodontal inflammation begins in response to oral pathogens by recruiting polymorphonuclear leukocytes (PMNs), which form a barrier to prevent the migration of microbes into the deep tissues [6]. PMNs attempt to clear out pathogens with their distinct functions, such as chemotaxis, degranulation, extracellular trap formation, phagocytosis, and release of defensins, antimicrobial peptides and reactive oxygen species (ROS) [7]. As the lesion progresses, a dense and more complex inflammatory infiltrate forms, including macrophages, lymphocytes, and plasma cells [8]. The ecological changes caused by bacterial and host-derived products and the manipulation of the immune system by some pathogens such as *Porphyromonas gingivalis* lead to a dysbiotic relationship between subgingival bacteria and the host, resulting in chronic, excessive, and dysregulated inflammation and disruption of homeostasis [9].

## 2. Oxidative Stress in Pathogenesis of Periodontal Disease

Reactive oxygen species are crucial for host tissue metabolism, e.g., cell signaling, proliferation, and renewal. However, excessive accumulation of ROS in tissues causes damage to lipids, carbohydrates, proteins, and deoxyribonucleic acids, eventually triggering a chronic inflammatory process. Various autoimmune or inflammatory oral conditions have been associated with cellular damage as a result of increased oxidative stress, such as lichen planus, lichenoid reactions, leukoplakia, pemphigus vulgaris, recurrent aphthous ulcers, oral squamous cell carcinoma, and periodontitis [10,11,12,13,14]. Current evidence suggests a close relationship between the pathogenesis of periodontitis and oxidative stress caused by oxygen-derived free radicals. Biomarkers associated with oxidative stress are elevated in periodontitis compared to a healthy periodontium [14,15]. PMNs play a central role in this mechanism as first responders. These cells migrate into periodontal tissues by responding to mediators, such as interleukin (IL)-8 or interferon (INF)-α, and releasing large amounts of ROS, triggering a cytotoxic respiratory burst catalyzed by nicotinamide adenine dinucleotide phosphate (NADPH) oxidase [16]. Essentially, ROS are protective in nature; however, their overexpression is cytotoxic to host cells [15]. Indeed, periodontitis patients demonstrate a hyper-responsive PMN phenotype, and their blood plasma contains a considerable number of pro-inflammatory cytokines that stimulate the production of ROS by neutrophils [16].

In periodontal tissues, PMNs generate ROS by responding to Fcγ receptors, but also to nonopsonized pathogens, such as *Fusobacterium nucleatum* and *P. gingivalis* [17,18]. The release of superoxide by neutrophils in the presence or absence of these nonopsonized pathogens suggests that extracellular ROS production is constitutive in periodontitis, whereas intracellular ROS are produced as a secondary response to periodontal inflammation [18].

The oxidative stress caused by PMNs damages not only the anaerobic periodontal pathogens but also the neutrophils themselves and plays an important role in periodontal destruction [19]. Tissue damage caused by ROS can occur in several forms: Lipid peroxidation, DNA damage, protein damage, oxidation of protective enzymes, and stimulation of proinflammatory cytokines [20]. Active PMNs can severely injure gingival epithelial cells, while their oxidants and proteases together damage periodontal tissues [21]. In addition, bone remodeling gets disrupted due to the increase in the receptor activator of nuclear factor kappa B ligand (RANKL)/osteoprotegerin ratio due to ROS, leading to osteoclastogenesis and alveolar bone loss [15,22] (Figure 2). These hyper-active and hyper-reactive PMN phenotypes are thought to be primarily associated with disrupted redox potential, particularly impaired regulation of antioxidant mechanisms by nuclear factor erythroid-2-related factor 2 (Nrf2) [23]. Nrf2 is a basic cap-n-collar leucine zipper transcription factor that is crucial in maintaining the periodontal tissue homeostasis by regulating anti-oxidative response element genes [24].

## 3. Anti-Oxidative Mechanisms in Periodontal Tissues

In a healthy state, the activities of ROS are balanced by various antioxidant mechanisms. When the antioxidant capacity is overwhelmed, oxidative stress occurs. This shift in the balance between ROS and antioxidants can be caused either by increased production and activity of ROS or by a disturbance in antioxidant defenses [17]. Antioxidant concentrations in the GCF of periodontitis patients have been found to be reduced compared to healthy individuals [25,26].

Nrf2 and its inhibitor Keap1 are important regulators of antioxidant defense mechanisms. Nrf2 controls the gene transcription of many antioxidant enzymes such as catalase, superoxide dismutase, NADPH, heme oxygenase, and glutathione peroxidase, which have been widely studied as antioxidants in periodontitis [27,28]. The Nrf2 pathway is suppressed in severe periodontitis, which is responsible for the elevated PMN levels and associated high oxidative damage in periodontitis. Nrf2-/- mice exhibit low catalase production by PMNs and increased periodontal tissue breakdown, suggesting a central role in periodontal pathogenesis [28]. This is also consistent with the promotion of RANKL-induced activation of kinases and nuclear factor activated T cells in Nrf2 deficiency, leading to osteoclast differentiation and consequent severe alveolar bone loss [28,29]. Nuclear Nrf2 is more frequently present and heme oxygenase-1 expression and luciferase activity are increased in experimental periodontitis when compared to healthy individuals [30]. Moreover, apoptosis of periodontal ligament stem cells, which are highly efficient in periodontal repair, is increased by oxidative stress, which also induces the antioxidant effect of Nrf2. Overexpression of Nrf2 increases antioxidants, such as NADPH-quinone oxidoreductase 1, heme oxygenase, and γ-glutamylcysteine synthetase, and induces cell proliferation while limiting apoptosis [31]. This suggests the importance of Nrf2 in maintaining the integrity of periodontal ligament cells and the antioxidant capacity of these cells. However, the Nrf2 response in gingival keratinocytes does not appear to be induced by *P. gingivalis* lipopolysaccharide, although it is stimulated by nicotine-induced oxidative stress [32]. Therefore, other antioxidant pathways besides Nrf2, such as forkhead box-O or sirtuins, should be considered in the pathogenesis of periodontitis, although there are still many unknowns in oxidative damage and antioxidant mechanisms in periodontal tissues [33,34,35,36].

Anti-oxidative mechanisms in peri-implant diseases also need to be briefly mentioned, considering that peri-implantitis is an increasing oral health problem. Peri-implant diseases share many common threads with periodontal diseases, but also exhibit different histopathologic characteristics due to anatomical and functional differences [37]. A scarce number of studies demonstrated increased oxidative stress specified with myeloperoxidase levels and nitric oxide metabolism in peri-implant disease, resembling inflammation in natural teeth [38,39,40]. To the best of our knowledge, only one study evaluated Nrf2 levels in dental peri-implantitis, suggesting an increase in Parkinsonism-associated deglycase 7, which regulates Nrf2, and increased oxidative stress measured with 8-hydroxy-deoxyguanosine, although Nrf2 expression was not altered in peri-implantitis mucosa [41].

## 4. Probiotics and Postbiotics

According to the Food and Agriculture Organization (FAO), “probiotics are live microorganisms which when administered in adequate amounts confer a health benefit on the host”. According to this consensus definition, probiotics should be administered or consumed alive. Apart from their use for oral health, probiotics may have many beneficial properties, such as competing with pathogens for adhesion, producing antimicrobial components, contributing to immunomodulation, and strengthening mucosal barrier function [42].

Postbiotics are all components produced and released by microorganisms as a result of metabolic activities. The components of postbiotics are intracellular or extracellular soluble byproducts of metabolism, as well as cell wall and cell membrane components, released as a result of bacterial cell lysis [43] (Figure 3). The term postbiotics is still very current, but its use beyond probiotics has led to its use in many different areas, such as the food industry, human health, nutrition, functional foods, and pharmaceuticals [44]. Postbiotics are cell-produced metabolic cocktails and do not contain microorganisms, making them more stable and user-friendly than probiotics. Unlike probiotics, postbiotics have the advantage of being more stable and longer lasting, posing fewer risks than live probiotics in immunocompromised individuals, and not spreading antibiotic resistance factors. Finally, publications investigating the potential immunomodulatory, antioxidant, antimicrobial, antibiofilm, antioxidant, and anticancer effects of postbiotics are increasing exponentially in the literature [45].

## 5. Antioxidant Abilities of Probiotics and Postbiotics

Oxidative stress damages macromolecules, such as proteins and nucleic acids, by increasing the level of oxygen radicals in the cell. Organisms have enzymes that respond to ROS by neutralizing them and preventing the damage they cause. Superoxide dismutase (SOD), glutathione peroxidase (GPx), glutathione reductase (GR), and non-enzymatic antioxidants are the main responses that protect the organism from oxidative stress [46]. The use of antioxidants from biological sources is becoming increasingly popular as supportive antioxidants [47].

Probiotics can exert their potential antioxidant effects in several ways. One of these ways is that some lactic acid bacteria show the ability to chelate metal ions, although the mechanism of action is unknown [22,47]. Redox-active metals, such as copper (Cu), iron (Fe), and cobalt (Co), which lead to redox reactions, stimulate the formation of ROS [48]. Probiotics may also have their own antioxidant enzymes and promote the production of antioxidant enzymes in the host. For example, Kullisaar et al. [49] demonstrated the presence of the enzyme superoxide dismutase (SOD) in *Lactobacillus fermentum*. Bacterial SODs offer promising results in the treatment of Crohn’s disease in studies using mouse models [50]. The products produced by probiotics as a result of their metabolic activities, such as short-chain fatty acids, vitamins, and folic acid, may also exhibit antioxidant activity on their own. The antioxidants of probiotics and postbiotics are summarized in Table 1. Although there are many studies in the literature on the antioxidant capacity of probiotics, a general overview of their antioxidant properties is provided here.

## 6. Probiotics and Nrf2 Activation

It is also known that probiotics can activate various pathways, such as Nrf2-Keap1-ARE, NFκB, MAPK, and PKC, which are responsible for the response to oxidative stress in the host [47]. In the light of all this information, it is clear that probiotics have the potential to exert antioxidant activity directly or indirectly. The Nrf2 activating capabilities of probiotics are summarized in Table 2.

Probiotic lactic acid bacteria activate not only the Nrf2 pathway but also Nrf2-associated antioxidant enzymes such as heme oxygenase-1, catalase, and SOD [65,68].

## 7. Postbiotics and Nrf2 Activation

In this section, we review the effects of probiotic components on Nrf2 activation in different systems. Although the concept of postbiotics has not yet come to the forefront in many of the studies mentioned, we have considered cellular components and byproducts of probiotics as postbiotics because they now fall within the definition of postbiotics.

### 7.1. Exopolysaccharides (EPS)

Many probiotic bacteria are capable of producing exopolysaccharides. The best-known benefits of exopolysaccharides to the host are in enhancing immune modulation and barrier function [69]. Some EPS extracted from probiotics have the ability to scavenge radicals directly and chelate metals [70,71]. For example, exopolysaccharides from strain *L. rhamnosus* GG, one of the best-studied probiotic strains, reduce H_2_O_2_-induced oxidative damage in intestinal epithelial cells and enhance the antioxidant response by activating Kelch-like ECH-associated protein 1 (Keap1)/Nrf2 signaling pathways [72].

### 7.2. Short-Chain Fatty Acids (SCFA)

Short-chain fatty acids, such as propionate, acetate, and butyrate, are functional for epithelial integrity, barrier function, and modulation of gut microbiota composition and immune response [73]. Butyric acid is one of the best-studied short-chain fatty acids and can be defined as a postbiotic that can be produced by probiotic bacteria [74]. Butyrate, for example, can trigger the antioxidant glutathione, leading to an oxidative stress response in colon cells [75]. Butyrate can also induce Nrf2 by histone acetylation in the liver [76]. Although butyrate has beneficial effects primarily in the gastrointestinal tract, butyrate produced by butyrogenic bacteria in the oral cavity plays a key role in the progression of periodontitis. Butyrate also promotes a proinflammatory response in fibroblasts of the periodontal ligament. Therefore, the role of butyrate in oral health can be considered a double-edged sword [77].

### 7.3. Carotenoids

Some carotenoids have the ability to induce phase II detoxification enzymes via the antioxidant response element (ARE) and Nrf2 [78]. Probiotic bacteria, such as *L. plantarum* and *Enterococcus gilvus,* can produce carotenoids [79,80].

### 7.4. Bioconverted Metabolites as Postbiotics

The nature of postbiotics may vary depending on the content of the medium in which the producing bacteria grow. Various bioactive metabolites can be produced by the bioconversion capabilities of probiotics. Lee et al. [81] published an important review on the use of bioactive metabolites obtained by probiotics-mediated bioconversion as therapeutic agents in periodontitis. Based on the concept of bioconversion mentioned in this study, it is possible to include all metabolites that probiotics can produce in the category of postbiotics, depending on the diversity of culture media in which they are cultivated. Thanks to their rich enzymatic arsenal, probiotic bacteria are capable of biotransforming many sources added to the culture media. For example, probiotics are able to produce fatty acids with different structures from fatty acids added to the culture media, bioactive peptides from milk and soy milk, and various flavonoid and phenolic compounds [82,83,84,85]. For example, *L. rhamnosus* GG and *L. plantarum* 299 v strains can ferment phenolic compounds to ferulic acid and caffeic acid [86]. Ferulic acid can induce heme oxygenase-1 via activation of ERK and Nrf2 in lymphocytes [87]. Cell proliferation and healing in oral tissues can be promoted by ferulic acid [88]. Caffeic acid appears to be an effective chemoprotective agent against oxidative damage, as it has been shown to stimulate the expression of detoxification enzymes via the ERK/Nrf2 pathway [89]. The increase in total antioxidant status (TAS) and decrease in total oxidative status (TOS) levels in inflamed gingival tissue suggest that caffeic acid has antioxidant therapeutic potential in periodontitis [90]. A natural compound known as urolithin A (UA) is derived from gut bacteria by the process of digested ellagitannins (ETs) and ellagic acid (EA), polyphenols abundant in fruits, nuts, and pomegranates [91]. Urolithin can enhance barrier functions through activation of aryl hydrocarbon receptor (AhR)-Nrf2-dependent pathways. By reversing barrier dysfunction, it reduces colitis in preclinical models, in addition to its anti-inflammatory and antioxidant properties [92]. Catechin is another metabolite that can be produced by probiotics through the fermentation of polyphenols [93]. Catechin can protect the gastric mucosa from oxidative damage by activating Nrf2 [94]. Casein-derived antioxidant peptides produced by microbial proteases are also gaining popularity for their potential Nrf2 activation. Casein-derived peptides obtained by microbial proteases can act as Keap1-Nrf2 signaling activators [95].

It should be noted that the above postbiotics have not yet been supported by direct in vitro or preclinical studies to regulate the Nrf2 response in periodontitis. However, postbiotics that have been associated with the Nrf2 response in other studies are likely to be candidates for activating periodontitis-related antioxidant response mechanisms in the future.

## 8. Postbiotics as Nrf2 Activators in Periodontitis: Scientific Evidence

Currently, there is limited evidence that postbiotics can be used in the oral cavity as modulators of host response, inthe form of in vitro studies. Some *Lactobacillus* strains are able to metabolize linoleic acid into various isomeric intermediates, i.e., fatty acids. Among these fatty acids, 10-oxo-trans-11-octadecenoic acid (KetoC) mediates the antioxidant response via the Nfr2-ARE pathway. KetoC induces the expression of antioxidant-related genes in gingival epithelial cells and administration of KetoC reduces ROS levels by promoting phosphorylation of signal-regulated kinase (ERK), leading to nuclear translocation of Nrf2 [96]. Nrf2 regulates the expression of heme oxygenase-1 (HO-1), which has been extensively studied for its immunomodulatory and cryoprotective roles [97]. Vo et al. [98] showed that a *Bacillus subtilis*-derived surfactin has the potential to induce Nrf2 activation and HO-1 expression. Surfactin inhibited particulate-matter-induced VCAM-1 expression by increasing Nrf2-dependent HO-1 activation in human gingival fibroblasts.

## 9. Postbiotics as Nrf2 Activators and Their Potential Use as Antioxidants in Periodontitis—Future Perspective

Anti-infective periodontal treatment (nonsurgical periodontal treatment/initial treatment/cause-related therapy) conventionally aims to control inflammation by reducing the microbial burden and prevent re-colonization by removing deposits on hard surfaces and by instructing proper oral hygiene measures (Figure 4). Following anti-infective treatment, periodontal surgery can be implemented to regenerate lost tissues and/or to obtain physiological contours. Various current nonsurgical and surgical approaches in periodontal treatment yield similar outcomes, but aimed clinical endpoints of therapy are not fully achieved in a significant number of sites or individuals [99,100]. While mild–moderate periodontitis cases can be effectively treated with nonsurgical treatment, more advanced cases require the inclusion of periodontal surgical treatments into the treatment plan. During the last two decades, the benefit of adjunctive applications, such as the use of laser or photodynamic therapy or host-modifying agents (anti-inflammatories, matrix metalloproteinase inhibitors, bisphosphonates, anti-oxidants etc.), as part of periodontal therapy, has been discussed very widely [101,102]. In fact, host modulation has been widely studied, since our understanding of periodontal treatment is shifting towards controlling the infection by reducing inflammation [101]. Indeed, animal studies have demonstrated that oxidative stress can be decreased with certain agents, such as etanercept, *Stemodia maritima* L. extract, gliclazide, *Calendula officinalis,* and atorvastatin [103,104,105,106,107]. On the other hand, scientific evidence is yet scarce to consider many host modulators for daily clinical use [108].

According to the European Federation of Periodontology treatment guideline, the scientific evidence for the complementary use of lasers, photodynamic therapy, probiotics, doxycycline in subantimicrobial doses, local or systemic bisphosphonates or nonsteroidal anti-inflammatory drugs, and local administration of statin or metformin gels is currently insufficient [109]. Therefore, exploration of different complementary treatment models, such as systemic or local administration of postbiotics, might be beneficial. Hypothetically, postbiotics could be used as an adjunct to periodontal treatment, especially in patients who do not respond to treatment, to reduce the inflammatory burden and, indirectly, infection by reducing oxidative stress. In addition, they could modulate the host response and prevent disease progression or recurrence in susceptible individuals.

## 10. Conclusions

Postbiotics can be considered as candidate host-modifying agents, since they are thought to be more reliable than probiotics regarding their clinical safety. Various postbiotics have been proven to have the potential to activate the Nrf2 pathway, which, in return, can reduce oxidative damage and inflammatory burden in periodontitis. The adjunctive use of postbiotics as Nrf2 activators can be particularly beneficial in non-responsive or high-risk periodontitis patients.

## Figures and Tables

**Figure 1 biomolecules-12-00729-f001:**
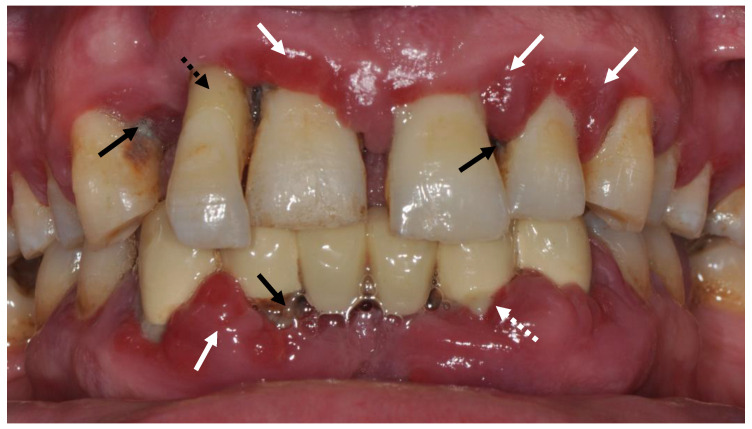
Clinical signs of severe periodontitis. White arrows: Signs of gingival inflammation (erythema, swelling and texture change). White dashed arrow: Suppuration. Black arrows: Dental deposits. Dashed black arrow: Signs of Gingival recession and pathologic tooth migration.

**Figure 2 biomolecules-12-00729-f002:**
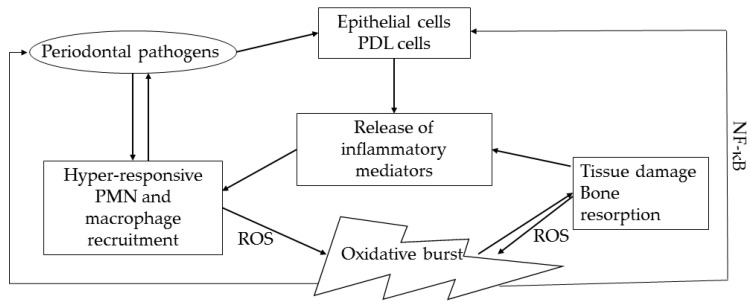
Oxidative stress in periodontal disease pathogenesis. PDL: Periodontal Ligament; ROS: Reactive Oxygen Species; NF-κB: Nuclear Factor kappa B.

**Figure 3 biomolecules-12-00729-f003:**
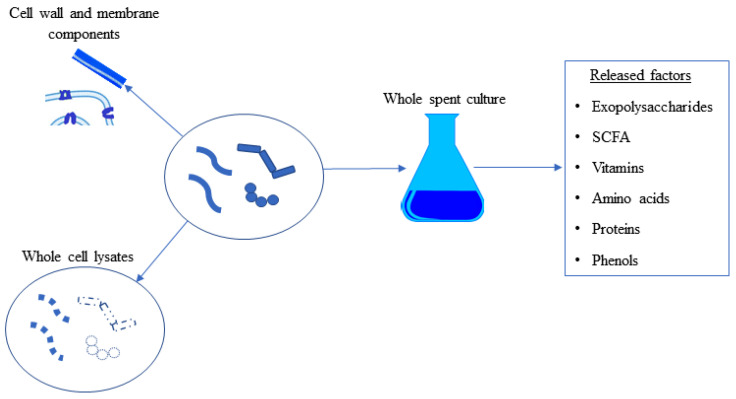
The contents that can be classified as postbiotics. Whole cell lysates include all intracellular and cell membrane/cell wall components. Only cell wall and membrane components can be used as postbiotics. Factors secreted into the extracellular environment, fermentation broth, or spent culture media can be considered postbiotics. SCFA: Short-Chain Fatty Acids.

**Figure 4 biomolecules-12-00729-f004:**
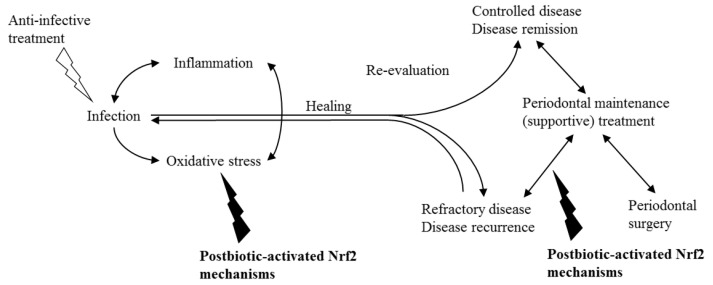
The primary aim of conventional periodontal treatment is to control inflammation and oxidative stress by reducing the infection. Yet, host-modifying agents can be beneficial in unresponsive and/or susceptible individuals. Postbiotic-activated Nrf2 mechanisms can be considered as potential agents to control the inflammation by reducing oxidative stress as an adjunct to anti-infective periodontal treatment or during supportive therapy to prevent recurrence in high-risk individuals.

**Table 1 biomolecules-12-00729-t001:** Probiotic bacteria and their postbiotic mediators as antioxidant agents.

Probiotics	Antioxidant Source	Mode of Action	Reference
*Bifidobacterium*, *Lactobacillus*, *Lactococcus*, *Streptococcus thermophilus*	Intact cells and spent culture media *	Scavenging activity on radicals	[51]
*Lactobacillus bulgaricus*, *Streptococcus thermophilus*, *Bifidobacterium lactis* Bb12 and *Lactobacillus acidophilus* La5 strains	Yogurt including indicated probiotics	Increased erythrocyte superoxide dismutase and glutathione peroxidase activities and total antioxidant status in Type 2 diabetes	[52]
*Bacillus coagulans* RK-02	*B. coagulans* derived exopolysaccharides *	Superoxide radical scavenging activity	[53]
*Lactobacillus rhamnosus GG, Lactobacillus retueria* (ATCC 20016), *Bifidobacterium breve* (ATCC 15700), *Probionebacterium freudenreichii* ssp.	Cell free culture extract *	DPPH scavenging activity	[54]
*Saccharomyces cerevisiae* IFST062013	Cell extract and autolysate *	Hydroxyl radical and nitric oxide scavencing activity	[55]
Human originated Lactobacilli and Bifidobacteria strains	Living cells	DPPH and ABTS scavenging activity	[56]
*Lactobacillus acidophilus* LA-14, *Lactobacillus casei* LC-11, *Lactococcus lactis* LL-23, *Bifidobacterium bifidum* BB-06, and *Bifidobacterium lactis* BL-4	Commercial probiotic Danisco^®^	Increased activity of glutathione peroxidase	[57]
*Lactobacillus plantarum* 200655	Living cells isolated from kimchi	DPPH and ABTS scavenging activity	[58]
SLAB51 Probiotic Formulation	Living cells	Activates SIRT1 pathway promoting antioxidant effects	[59]

* can be considered as postbiotics. ATCC: American Type Culture Collection. DPPH: 2,2-diphenyl-1-picrylhydrazyl. ABTS: 2,2’-azino-bis(3-ethylbenzothiazoline-6-sulfonic acid). SIRT1: Sirtuin I.

**Table 2 biomolecules-12-00729-t002:** Nrf2 activating abilities of probiotic bacteria.

Source	Antioxidant Activity	Reference
*Lactobacillus plantarum*	Increased production of caffeic acid by *L. plantarum* in combination with apple juice promotes Nrf2 activation, resulting in improved diastolic function in chronic ischemic myocardium.	[60]
Commercial probiotic	Probiotics combined with *Illicium verum* extract and glucose oxidase enzyme upregulate hepatic and jejunal Nrf2/Keap1 pathway.	[61]
Commercial probiotic	Probiotic supplement improves antioxidant defence of cardiomyocytes by regulating Nrf2 and caspase3 gene expression in type 2 diabetic rats.	[62]
*Lactobacillus rhamnosus* GG and its spent culture supernatant	Improvement of myocardial dysfunction in obese mice exposed to intermittent hypoxia by activating Nrf2 Pathway	[63]
*Lactobacillus rhamnosus* GG	Significant induction of Nrf2 target transcripts in liver tissue via situmulation of xenobiotic	[64]
*Lactobacillus plantarum* NA136	Increased the nuclear translocation of Nrf2 leading to improved antioxidant response in case of non-alcoholic fatty liver disease	[65]
Gut-resident Lactobacilli	Activation of Nrf2 response against oxidative liver injury through gut-liver axis	[66]
MIYAIRI 588–a butyrate-producing probiotic strain	Reduced nonalcoholic fatty liver disease progression via Nrf2 pathway	[67]

Keap1: Kelch-like ECH-associated protein 1.

## Data Availability

The study did not report any data.

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
