# Peer review of "Targeting Nrf2 with Probiotics and Postbiotics in the Treatment of Periodontitis"

_biomolecules, 2022, doi:10.3390/biom12050729_

Round 1

Reviewer 1 Report

Dear Authors, 

I congratulate the authors for writing this extensive narrative review. However, I would like to suggest a few areas which can be improvised. 

  1. Title - mention "the" before treatment of periodontitis.
  2. Discuss the role of Oxidative stress in the pathogenesis of various oral diseases including acute and chronic conditions. The intention is to showcase the involvement of Oxidative stress. Acute conditions such as recurrent aphthous ulcer (DOI: 10.1007/s00784-013-1181-2) potentially malignant disorders such as oral leukoplakia (PMID: 30804105), chronic malignant conditions such as oral cancer (PMID: 27076834) 
  3. Authors can add the full versions of the abbreviations used in Tables 1 & 2 as footnotes. 
  4. In section 8, the authors have mentioned few in vitro and animal studies have been carried out. Please cite the references immediately after this sentence. Also, in the coming paragraphs, I could not find any description of animal study. 
  5. I encourage authors to mention the merits and demerits of conventional methods available for the treatment of periodontal diseases, such as local drug delivery, and SRP, as authors are proposing to replace the conventional methods with probiotic and postbiotic. 
  6. Modify the legend of Figure 2. It looks abrupt. 

Best Wishes 

Author Response

To begin with, authors express their gratitude and appreciation for your contribution and comments regarding the study.

1) Title - mention "the" before treatment of periodontitis.

It was added. Please see the title.

2) Discuss the role of Oxidative stress in the pathogenesis of various oral diseases including acute and chronic conditions. The intention is to showcase the involvement of Oxidative stress. Acute conditions such as recurrent aphthous ulcer (DOI: 10.1007/s00784-013-1181-2) potentially malignant disorders such as oral leukoplakia (PMID: 30804105), chronic malignant conditions such as oral cancer (PMID: 27076834)

A paragraph is included into the first paragraph of the Section 2. Please see the lines 79-85 and 114-116.

3) Authors can add the full versions of the abbreviations used in Tables 1 & 2 as footnotes.

All abbreviations were considered and added to tables as footnotes. Please see Tables 1 and 2.

4) In section 8, the authors have mentioned few in vitro and animal studies have been carried out. Please cite the references immediately after this sentence. Also, in the coming paragraphs, I could not find any description of animal study.

We just would have emphasized in vitro studies. The misunderstanding was omitted. Please see the section 8. Please see the line 294.

5) I encourage authors to mention the merits and demerits of conventional methods available for the treatment of periodontal diseases, such as local drug delivery, and SRP, as authors are proposing to replace the conventional methods with probiotic and postbiotic.

We added a discussion on the current evidence regarding conventional and adjunctive periodontal treatment methods to the Section 9. To clarify, the mechanical debridement is essential in periodontal treatment, and what we propose is the adjunctive use of postbiotics rather than replacing the conventional approach. Please see the lines 313-327, 335-340.

6) Modify the legend of Figure 2. It looks abrupt.

It was modified- Please see Figure 2.

Reviewer 2 Report

The work is well written and deals with a very interesting topic. The aim of this narrative review is to describe the role of Nrf2 in periodontal health and as possible target of periodontal treatment, by means of probiotic and postbiotics.

The section 1 is a kind of general introduction to the topic, I would suggest to add a paragraph to contextualize the topic and a further paragraph to describe and strengthen the aim of the review, briefly explaining why focusing on Nrf2 .

The quality of figure 1 is low: it is difficult for example observe suppuration. I would suggest to upload a better clinical image.

Section 2 (Oxidative stress in periodontal disease pathogenesis) would benefit of an additional figure. Moreover, since Nrf2 is a core element in the review, I suggest to add at the end of this section a brief sentence to provide an overview of its biological role and function.

Author Response

To begin with, authors express their gratitude and appreciation for reviewer’s contribution and comments regarding the study.

1) The section 1 is a kind of general introduction to the topic, I would suggest to add a paragraph to contextualize the topic and a further paragraph to describe and strengthen the aim of the review, briefly explaining why focusing on Nrf2.

A paragraph briefly describing the aim of this narrative review was added to the very beginning of the manuscript. Please see the lines 29-40.

2) The quality of figure 1 is low: it is difficult for example observe suppuration. I would suggest to upload a better clinical image.

Please see the replaced figure. We apologize for the initial figure with the decreased image quality, which seems to be due to a technical problem when uploading the file. We are eager to upload a separate file of the same figure or change the image with a different clinical periodontitis image, in case the problem persists.

3) Section 2 (Oxidative stress in periodontal disease pathogenesis) would benefit of an additional figure. Moreover, since Nrf2 is a core element in the review, I suggest to add at the end of this section a brief sentence to provide an overview of its biological role and function.

A brief sentence regarding the role and function of Nrf2 was added to the end of the Section 2. The functions of Nrf2 in periodontal tissues are discussed in more detail within the Section 3. Finaly, an additional figure was added. Please see the lines 114-116.